# MQTTset, a New Dataset for Machine Learning Techniques on MQTT

**DOI:** 10.3390/s20226578

**Published:** 2020-11-18

**Authors:** Ivan Vaccari, Giovanni Chiola, Maurizio Aiello, Maurizio Mongelli, Enrico Cambiaso

**Affiliations:** 1Consiglio Nazionale delle Ricerche (CNR), IEIIT Institute, 16149 Genoa, Italy; maurizio.aiello@ieiit.cnr.it (M.A.); maurizio.mongelli@ieiit.cnr.it (M.M.); enrico.cambiaso@ieiit.cnr.it (E.C.); 2Department of Informatics, Bioengineering, Robotics and System Engineering (DIBRIS), University of Genoa, 16145 Genoa, Italy; chiolag@acm.org

**Keywords:** Internet of Things, dataset, MQTT, machine learning, detection system, artificial intelligence

## Abstract

IoT networks are increasingly popular nowadays to monitor critical environments of different nature, significantly increasing the amount of data exchanged. Due to the huge number of connected IoT devices, security of such networks and devices is therefore a critical issue. Detection systems assume a crucial role in the cyber-security field: based on innovative algorithms such as machine learning, they are able to identify or predict cyber-attacks, hence to protect the underlying system. Nevertheless, specific datasets are required to train detection models. In this work we present MQTTset, a dataset focused on the MQTT protocol, widely adopted in IoT networks. We present the creation of the dataset, also validating it through the definition of a hypothetical detection system, by combining the legitimate dataset with cyber-attacks against the MQTT network. Obtained results demonstrate how MQTTset can be used to train machine learning models to implement detection systems able to protect IoT contexts.

## 1. Introduction

The volume of data exchanged through global networks is increasing every year, due to the huge number of devices connected to ICT networks. Moreover, the rapid expansion of the Internet of Things (IoT) phenomenon is considered a key factor of this high number of traffic volume [1]. Thanks to IoT, simple objects gain the ability to process and exchange information among themselves or other entities. If we consider the nature of the applications on the Internet, the value of exchanged information is considerable, as exchanged data are often sensitive and contain relevant information. Such aspect is especially relevant in the IoT context, where location and nature of devices make them exchange sensitive information on the network. In virtue of this, security of IoT environments is a critical point and IoT systems must be secured to be able to transmit data through the Internet freely, without being affected by cyber-attacks.

In order to protect the data exchanged in ICT networks, including IoT environments, detection and mitigation system are employed, to counter cyber-threats. In this context, thanks to the rapid growth of machine learning (ML) and artificial intelligence (AI) algorithms, networks monitoring and prediction of incoming cyber-attacks is nowadays possible [2,3]. Nevertheless, it is well-known that ML and AI systems require a large amount of well-structured data to be adopted, in order to train models used to identify malicious situations [4]. By focusing for instance on IoT context, communication traffic of IoT environments can be used by ML/AI algorithms to train a detection algorithm to identify running attacks on the network.

Notwithstanding, the field of datasets used in the IoT context is extremely limited. In particular, general datasets used in cyber-security (e.g., KDDCUP99 [5], UNSW-NB15 [6], or NIMS [7]) are used, although they are rarely suitable to IoT environments, due to the limited support to dedicated protocols used in IoT networks.

Starting from the problem of the few available datasets available in the IoT context, in this paper we introduce MQTTset, a novel dataset focused on IoT. In particular, MQTTset includes communications based on the Message Queue Telemetry Transport (MQTT) protocol, a publish/subscribe protocol introduced in 1999 [8] and considered an IoT standard protocol by the OASIS group [9]. Although it’s designed to be used in IoT environments, MQTT is even adopted for applications external to IoT like mobile health monitoring or push notification services [10,11,12]. MQTTset is composed of IoT devices of different nature (e.g., temperature, humidity, motion sensors, etc.), in order to simulate a smart home/office/building environment. In addition, MQTTset includes both legitimate and malicious traffics. Hence, it is potentially possible to use MQTTset to train ML/AI models in order to characterize the legitimate behavior and identify malicious situations. In this paper, we first propose the MQTTset dataset, hence validate it by adopting known ML/AI algorithms in order to characterize legitimate traffic and identify potential threats on the network. Finally, in order to help the research community to investigate the growing IoT context, we publicly released the MQTTset dataset, including both legitimate and attack traffics, expressed in form of PCAP packet capture files.

The remaining of the paper is structured as follows: Section 2 reports the related work on the topic. Section 3 describes in detail the created dataset and the validation activities. Section 4 presents the obtained results. Finally, Section 5 concludes the paper and reports further works on the topic.

## 2. Related Work

Considering the IoT security topic, several attacks against IoT networks are found in literature, starting from the evaluation of the impact of well-know attacks applied to IoT environments [13], up to the proposal of novel threats against IoT networks, protocols or nodes [14,15,16]. Protection of IoT networks and systems from cyber-threats is an open research challenge, due to the constant appearance of novel threats targeting such platforms.

In these years, machine learning algorithms are adopted to detect cyber-attacks against infrastructure and networks. Particularly, deep learning approaches are adopted to detect cyber-attacks by training the algorithm with the KDDCUP99 [17], while random forest, decision tree and gradient boost algorithms are adopted to implement intrusion detection system with KDDCUP99 [18,19,20] and naïve bayes algorithms are adopted for cyber-protection in [21]. In order to design and validate efficient and accurate protection systems to detect ICT attacks, the availability of public datasets is a critical point in the research world. By analyzing datasets considered in literature, although published in 1999, the KDDCUP99 dataset is still adopted to implement detection systems by comparing different machine learning algorithms [22], or by implementing specific algorithms such as random forest to classify network traffic flows [23]. Notwithstanding, although KDDCUP99 is widely adopted in cyber-security [24,25], it is not a good choice to adopt it in IoT scenarios, since it is not intended to be used in this context, as it includes attacks on conventional ICT networks that are difficult to adapt to IoT environments.

If we consider instead datasets used for detection of attacks against IoT, UNSW-NB15 and NIMS are combined in [26] with simulated IoT sensors, in order to identify running attacks. Although authors consider HTTP, DNS and MQTT protocols, the MQTT traffic generated by the IoT sensors is not publicly available, while the other traffics are. Ref. [27] makes instead use of a custom dataset created by combining commercial IoT solutions (like Echo Dot, Belkin NetCam, Hive Hub, Samsung Smart Things Hub) with different communication protocols (Wi-Fi, ZigBee and Bluetooth Low Energy) to classify cyber-attacks. Ref. [28] combines instead simple IoT sensors, like temperature, motion, air pressure sensors, to validate machine learning algorithms (one-class classification, Isolation forest (iForest), Local Outlier Factor (LOF)), while Ref. [29] focuses on a real IoT industrial scenario by designing an intrusion detection system based on machine learning to detect cyber-attacks against an industrial networks. Another dataset adopted to compare the effectiveness of classification algorithms on IoT malware infections and IoT benign traffic by using IoT-23 dataset is investigated in [30]. This dataset contains DNS traffic focused on Mirai, Torii, IoT Trojan, Kenjiro, Okiru, Haji me and other botnet. N-BaIoT is another dataset used to detection and mitigate botnet attacks in the IoT context [31] focused on Wi-Fi communication. Although the adopted datasets are in this case particularly interesting and variegate, authors did not release them publicly. Hence, the possibility to exploit them for research purposes is extremely limited.

By focusing specifically on MQTT, different research works implement detection algorithms based on MQTT dataset. In this context, a variant of the KDDCUP, called NLS-KDD, is adopted in [32] to implement an artificial neural network (ANN) able to prevent attacks against MQTT. Nevertheless, by analysing the adopted dataset, it does not include MQTT application data, by focusing just on TCP transport layer packets. Therefore, in this scenario, a specific MQTT attack may not be detected, if exploiting the application layer protocol. Ref. [33] builds instead an MQTT dataset for detection approach based on machine learning called TON_IoT. Although publicly available, TON_IoT does not include all MQTT packets send/received during a connection: in particular, the authentication phase for both MQTT and TCP, involving the sensor and the broker, is not found in the dataset. Similarly, disconnections are not present. As authentication and disconnection phases are a critical aspects of IoT devices’ communications, the dataset is considered incomplete. In addition, the TON_IoT dataset includes a single TCP connection for all the nodes, hence making it particularly difficult to distinguish different nodes, for instance at transport layer. MedBIoT is another dataset related to IoT botnet focused on the detection of botnet attacks as Mirai, Yakuza and Torii [34] but in this dataset the authentication phase is not present. Another IoT dataset is BoT-IoT [35]. Such dataset has been adopted for different applications, such as to train deep learning based intrusion detection systems [36] or to train a C5 classifier and a One Class Support Vector Machine classifier to detect cyber-threats on the network [37]. In BoT-IoT, MQTT is exploited for communications with AWS services. Nevertheless, the raw PCAP traffic data related to MQTT was not released. As previously anticipated, considering previous works on the topic, different custom datasets are build and adopted to design protection systems, by also considering the MQTT protocol. Particularly, in literature, data of DHT11 temperature sensors connected to MQTT public services are adopted [38], as well as temperature, inRow and coolant sensors [39]. Although such works are promising, related datasets are not publicly available. Instead, concerning datasets of devices such as sensors and actuators proposed in [40], they have been publicly released, although not representative of a real network. In particular, in this case, data regarding the authentication phase are missing. In addition, raw PCAP files are also not available.

If we consider the adoption of deep learning methods for cyber-protection purposes, Ref. [41] analyses and reports the most adopted datasets in this context. Particularly, as author reported, the only dataset available including MQTT traffic is BoT-IoT [35], although, as mentioned before, it does not include raw and extracted network data related to MQTT.

A summary of the dataset available and their missing aspects, compared to the proposed work, is reported in Table 1.

Unlike presented related works on the topic, in this paper we introduce MQTTset, a dataset including raw traffic data related to the MQTT protocol, widely adopted in IoT environments. It is important to consider that many datasets publicly available, like KDDCUP99, are released as a set of comma separated values (CSV) files. Hence, a pre-processing of raw data is accomplished, before releasing the dataset. Instead, MQTTset is released both as CSV and PCAP raw data, in order to let the possibility to manually process raw information and produce different CSV files, according to the need. Particularly, the proposed dataset contains both legitimate and attack traffic, by considering all the data of the communications on the reference scenario (e.g., authentications, disconnection, etc.). We also validate the possibility to identify anomalous traffics by using the dataset to train ML/AI algorithms, for anomaly detection purposes. Finally, in order to help the scientific community to investigate the IoT topic and to adopt ML/AI algorithms on the presented dataset, in order to identify running threats, we publicly released the proposed MQTTset dataset.

## 3. MQTTset Dataset

This work aims to create MQTTset, an IoT dataset focused on MQTT communications. MQTTset was built by using IoT-Flock [42], a network traffic generator tool able to emulate IoT devices and networks based on MQTT and CoAP protocols. IoT-Flock provides the ability to configure the network scenario, in terms of nodes (e.g., sensor type, IP addresses, listening ports, etc.) and communications (e.g., time interval used for communications between the sensors and the broker). In addition, the tool implements different cyber-threats against the MQTT and CoAP: publish flood, packet crafting attacks, segmentation fault attack against CoAP (making use of a null Uri-path), and memory leak attacks against CoAP (by using invalid CoAP options during packets forging).

In order to create a dataset representative of a real network, in our scenario, we deployed different IoT sensors connected to an MQTT broker. Particularly, such broker is based on Eclipse Mosquitto v1.6.2, while the network is composed by 8 sensors. The considered scenario can be assimilated to a smart home environment, where sensors, uniquely identified by an IP address, retrieve information like temperature, light intensity, humidity, CO-Gas, motion, smoke, door opening/closure and fan status at different temporal intervals. According to Figure 1, sensors are located into two separated rooms.

The sensors network is implemented in a limited access area (both physically and virtually) where sensors communicate with the broker. In the network, no additional components (e.g., firewall) are installed. Indeed, the traffic is captured from the broker itself. Instead, during the attack phases, the malicious node is directly connected to the broker in order to execute the cyber-attacks. The position of the attacker node inside the network is not relevant since its aim is to attack the MQTT broker due to the nature of the selected attacks.

Each sensor is configured to trigger communication at a specific time, depending on the nature of the sensor. For instance, a temperature sensor may send information on the measured temperature on the environment at predefined time intervals, e.g., every hour. Instead, a motion sensor communicates on the network only when a movement is detected. Hence, in this case, since a “periodic” communication is not suitable, a “random” one was adopted, by simulating motions at random times. The communication behaviour is reported in Table 2 in the *type* column, where *periodic* indicates the sending of a periodic message (sent every *n* seconds, with *n* reported in the *messages frequency* column) and *random* indicates that sending is accomplished at random periods, every a random *m* value, with m≤n, with *n* defined previously. By analyzing communication aspects, the dataset simulate a real behavior of a home automation since sensors communicate based on their functionality.

Each sensor is set up with a data profile and a topic used by the MQTT broker. The data profile consists of the type of data used by the sensors, such as the ranges used by temperature or humidity sensors, or the commands adopted by door lock sensors. Instead, the topic is the identifier of the channel used to publish or receive information. In our scenario, the MQTT broker, identified by the IP address 10.16.100.73, is listening on plain text port 1883. Information on all the involved sensors are shown in Table 2. Furthermore, some of the sensors connected to the network, in addition to sending information, also have subscriber functions, to retrieve the data exchanged on the network.

The MQTTset dataset includes network traffic related to MQTT version 3.1.1. Authentication is not enabled, hence, no username and password exchange is required to authenticate clients to the broker. In addition, only plain text communications are included. Although this may represent a limit of the proposed dataset, it provides packets inspection capabilities [43], as well as the possibility to consider a wider set of parameters included in network packets.

The generated MQTT traffic is represented as a packet capture (PCAP) file, captured during the generation of MQTTset data. Capture time refers to a temporal window of one week (from Friday at 11:40 to Friday at 11:45). The dataset is publicly available (More information are available in Section 6) and it is represented by 11,915,716 network packets and an overall size of 1,093,676,216 bytes.

Starting from MQTTset, several possible intrusion detection and traffic characterization applications related to the MQTT protocol may be implemented. Particularly, as previously mentioned, to the best of our knowledge, a comprehensive and publicly available dataset focused on IoT protocols like MQTT is missing. In addition, MQTTset includes not only legitimate traffic, but also malicious one, we will now briefly introduce the considered threats. In this scenario, we integrated popular and easy to detect cyber-attacks against MQTT but it is possible to integrate more complex attacks such as zero-day [37] or innovative attacks against MQTT such as SlowITe [14] which is characterized by a particularly low attack band since it is a slow dos attack, the computational capabilities and bandwidth required to perform this attack are very low making it difficult to identify and mitigate. Being publicly accessible, researchers will be able to integrate their attacks with the dataset for analysis/detection/mitigation purposes. Referring to such attacks, we will now briefly introduce them.

### 3.1. Considered Cyber-Attacks

As previously anticipated, MQTTset includes real attacks implemented to target the considered MQTT network, in order to include in the dataset additional PCAP files which could be adopted, for instance, to validate detection algorithms. Particularly, the following attacks are part of MQTTset and summarized in Table 3.

#### 3.1.1. Flooding Denial of Service

Denial of service attacks are executed to prevent the service to serve legitimate clients [44]. In this case, the MQTT protocol is targeted with the aim to saturate the broker, by establishing several connections with the broker and sending, for each connection, the higher number of messages possible. In order to implement this attack, we adopted the the MQTT-malaria tool [38], usually adopted to test scalability and load of MQTT services.

#### 3.1.2. MQTT Publish Flood

In this case, a malicious IoT device periodically sends a huge amount of malicious MQTT data, in order to seize all resources of the server, in terms of connection slots, networks or other resources that are allocated in limited amount. Differently on the previous attack, this attack tries to saturate the resources by using a single connection instead of instantiate multiple connections. This attack was generated in this case by using a module inside the IoT-Flock tool [42].

#### 3.1.3. SlowITe

The Slow DoS against Internet of Things Environments (SlowITe) attack is a novel denial of service threat targeting the MQTT application protocol [14]. Particularly, unlike previous threats, being a Slow DoS Attack, SlowITe requires minimum bandwidth and resources to attack an MQTT service [45,46,47]. Particularly, SlowITe initiates a large amount of connections with the MQTT broker, in order to seize all available connections simultaneously. Under these circumstances the denial of service status would be reached.

#### 3.1.4. Malformed Data

A malformed data attack aims to generate and send to the broker several malformed packets, trying to raise exceptions on the targeted service [48]. Considering MQTTset, in order to perpetrate a malformed data attack, we adopted the MQTTSA tool [49], sending a sequence of malformed CONNECT or PUBLISH packets to the victim in order to raise exceptions on the MQTT broker.

#### 3.1.5. Brute Force Authentication

A brute force attack consists in running possible attempts to retrieve users credentials used by MQTT [50]. Regarding MQTTset, the attacker’s aim is to crack users’ credentials (username and password) adopted during the authentication phase. Also in this case, we used the MQTTSA tool [49]. Particularly, in order to recall to a real scenario, we adopted the *rockyou.txt* word list, that is considered a popular list, widely adopted for brute force and cracking attacks [51]. For our tests, the credentials are stored on the word list used by the attacker.

### 3.2. MQTTset Validation

After we defined the sensors included in the network and generated the dataset, we decided to use MQTTset to provide a publicly available dataset to be used for detection purposes. As mentioned above, MQTTset embeds IoT related traffic, in particular, MQTT communications. In order to validate MQTTset, we designed an intrusion detection system, hence applied on the dataset, combining legitimate MQTT traffic with different cyber-attacks (mentioned above) targeting the MQTT broker of the network. Both legitimate and attack traffics are part of MQTTset. Subsequently, the different datasets referring to legitimate and malicious situations were mixed together and used to carry out training and prediction of our algorithms, to validate the possibility to use MQTTset to test and implement a novel intrusion detection algorithm.

For validation of potential intrusion detection systems, we considered the following algorithms: neural network [52], random forest [53], naïve bayes [54], decision tree [55], gradient boost [56] and multilayer perceptron [57]. In each case, a data pre-processing phase is carried out, with the aim of extracting the necessary features able to characterize anomalous, hence attack, traffics/connections. This phase is extremely crucial, as, depending on the selected features, the adopted algorithm may lead to different results. Moreover, the selected features have to be picked up accurately, as they have to represent and characterize a specific category of network traffic [3].

All available features able of describing a connection were then recovered directly from the raw network data. Features are extracted and filtered in order to focus on the most relevant ones able to characterize potential attacks and our legitimate traffic. In particular, the features removed are:Source/destination addresses and ports: such features are removed in order to allow detection to be more independent on networking configuration details (useful for DoS/DDoS attacks)Communication times: such features are removed since the identification of attacks must not be dependent on times or schedulestcp.stream: such parameter are related to a single execution, not useful for detectiontcp.checksum: it is a unique value for each packet of the communicationMQTT clientId, password, username and related lengths: such parameters are related to a single execution and configuration and can be easily altered by an attackerMQTT topic: such parameter, easy to tune by the attacker, could be adopted to discriminate legitimate and malicious behaviouriRTT: a dedicated parameter used from Wireshark to define time between packets, not related to a connectiontcp.window_size_value: it is a parameter related to a single packet

The full list of selected features extrapolated and provided by MQTTset is reported in Table 4. Such features were extracted both for the legitimate and the malicious cases.

A summary of the workflow of the proposed work is reported in Figure 2: starting from raw network traffics provided by MQTTset, features extraction is accomplished. Hence, data are combined to mix legitimate and malicious traffics. Since the features extracted are time-independent, the mix of legitimate and malicious traffic is executed with a random approach but with a fixed seed (with value 7) in order to replicate the dataset easily. On the mixed traffic generated, we adopt different detection algorithms, with the aim to identify anomalies on the generated traffic data.

Obtained results will be presented and discussed in the next section.

## 4. Testbed and Obtained Results

After pre-processing and features extraction stages has been accomplished, and data are combined/mixed to generate a single dataset including both legitimate and malicious traffic data, the aim is now to validate all the intrusion detection algorithms we selected above. Selected algorithms are implemented in Python programming language, by using well known machine learning and artificial intelligence libraries and tools such as Sklearn [58], Tensorflow [59] and Keras [60]. All the algorithms have been tested on the same (mixed) dataset generated and on the same host (in details, a MacBook pro 2017 with a 2.5 GHz Intel Core i7 dual-core, 16 GB of RAM and 512 GB SSD disk), to avoid potential deviations referred to hardware or data changes. In this way, we keep consistency on tests and results. Since the MQTTset is composed of different types of MQTT traffic (see Section 3), the detection system must solve a multi-classification problem as it must not only identify an attack but, based on the training phase of the system, also predict the nature and type of attack. The multi-classification approach is considered since in a real scenario, a system could be target for cyber-attacks with different behaviour, payload and characteristics. A detail detection could be important in order to mitigate efficiently the identified threats. Based on this real scenario, an intrusion detection system should be able to identify malicious behaviours, in order to protect the system from attacks. For these reasons, we have implemented and validated intrusion detection algorithms able to make multi-class predictions [61], as our aim is to classify multiple threats. The decision tree (DecisionTreeClassifier) is implemented with *gini* criterion, *best* splitter and maximum depth set until all leaves are pure, while the random forest classifier (RandomForestClassifier) is tested with the default configuration. Instead, the gradient boost (GradientBoostingClassifier) is configured with maximum 20 estimator. Regarding the deep learning approach, the multilayer perceptron (MLPClassifier) is configured with a max iteration set to 130, a batch size to 1000, an activation function set to *relu* and with *adam* solver. The neural network, instead, is implemented by using the sequential algorithm with Kers (Sequential) with the first hidden layer consisting of 50 nodes, the second of 30, the third of 20 and finally the last with 6 nodes relating to the 6 classes. The hidden layer are characterized by a *relu* activation function and a *normal* kernel initializer except the last hidden layer since it is set with a *softmax* activation function. Finally, the naive bayes approach is configured by using a Gaussian Naive Bayes (GaussianNB). In order to replicate the tests, we have set a seed with a value of 7 adopted in the algorithms.

In order to test the selected intrusion detection algorithms, the dataset has to be splitted into two parts: training (70% of traffic data, in terms of generated records) and test (the remaining 30% of traffic data). Hence, as for other similar approaches [62], after training is accomplished, the test phase is perpetrated. Table 5 shows the results obtained for each of the selected algorithms, in terms of accuracy, F1 score [63] and execution time.

Accuracy is the ratio of number of correct predictions to the total number of input samples. Instead, the F1 score is the harmonic mean of precision and recall, where precision is the number of true positives divided by the number of all positive results, while recall is the number of true positives divided by the number of all tests that should have been positive (i.e., true positives plus false negatives) [64].

By analyzing in detail obtained results, by focusing on artificial neural network algorithms, the neural network provided an accuracy of 0.993 with an F1 score equal to 0.993, while the multilayer percetron resulted in 0.946 of accuracy and 0.963 of F1 score. Instead, by analyzing decision tree algorithms, random forest results gave us an accuracy value equal to 0.994 with an F1 of 0.994, while decision tree analysis reported an accuracy equal to 0.977 and an F1 score equal to 0.985 and the gradient boost obtained both accuracy and F1 score near to 0.991. Finally, for the part of supervised learning, the naïve bayes obtained accuracy of 0.987 and F1 score of 0.989. In order to better analyze the results, the confusion matrix are calculated and reported in Table 6, Table 7, Table 8, Table 9, Table 10 and Table 11.

All the algorithms have obtained an accuracy level above 98%, while the F1 score is found to always be above 97%. On the basis of the confusion matrices, the multilayer perceptron classifies most traffic well while gradient boost is the best for classifying legitimate traffic. In particular, on the basis of all the matrices and algorithms, flood, malformed data and SlowITe attacks are complex to identify, since most of the times they are classified as a different scenario.

Despite the difficulty of the algorithms to classify attacks properly, the accuracy and F1 score values are high. This consideration is due to the number of records relating to legitimate traffic, since it is composed by a much greater number of records than the sum of the records of all malicious traffics. In fact, the order size of legitimate traffic is in the billion while that of malicious traffic is in the order of thousands. In particular, the sum of malicious traffics is 165,281 and the legitimate traffic is 11,915,716. More details are available in Section 3.

Therefore, data related to legitimate traffic greatly influence the calculation of the metrics. Based on this concept, a balancing about legitimate and malicious dataset is needed, in order to calculate more precise accuracy and F1 score metrics.

### Additional Tests

In order to implement a more balanced dataset, we elaborated the dimensions of the single datasets going to balance the reports since, as shown in Section 3, the legitimate traffic was much larger than the sum of the malicious traffic and the relative results were influenced by this value. For this reason, we have revisited the size of the individual traffic data related to the malicious scenarios, by replicating each threat, in order to have a final size in the same order of magnitude of the data related to the legitimate scenario. In particular, we have set the size of the dataset at ten million records and increased each single traffic to two million records (as we have five traffic scenarios). In this way, the sum of the malicious traffic is balanced with the legitimate traffic. Once we created this extended dataset, we ran the same algorithms and calculated accuracy and F1 score. The results obtained are reported in Table 12.

By comparing Table 5 and Table 12, obtained results are clearly different in terms of accuracy and F1 score. The algorithms have an accuracy and F1 score between 87% and 91%, except the naïve bayes algorithm, where the results are around 64% in accuracy and 68% in F1 score. In order to compare confusion matrices between balanced and unbalanced datasets, we report in the following the results obtained for random forest, neural network and naïve bayes confusion matrices in Table 13, Table 14 and Table 15, in order to report one matrix for each algorithm approach (artificial neural network, decision tree and probabilistic classifiers).

By analyzing the confusion matrices, it can be seen that datasets are balanced. In particular, such balancing ensures the values of the most real metrics and not influenced by a specific value. By analyzing the matrices in detail, it can be seen how the neural network correctly classifies legitimate traffic while the random forest identifies flood and malformed traffic and finally the naïve bayes algorithm classifies bruteforce attacks. Instead, all algorithms are able to identify the SlowITe attack very precisely. These results could be considered more precisely and accurate since the dataset is balanced in order to perform balanced tests on the dataset. Moreover, the algorithms shown some lack in terms of detection of the attacks since sometimes the classification process is not able to identify the correct traffic.

As we have shown, MQTTset provides the possibility to analyse MQTT traffic and to implement and validate intrusion detection systems algorithms able to detect threats targeting MQTT networks.

## 5. Conclusions and Future Works

In this work we presented MQTTset, a legitimate dataset related to the MQTT protocol, widely adopted in IoT networks. The dataset was built from a network of IoT sensors of different nature (temperature, motion sensor, humidity, door locker, etc.), able to communicate on the network in order to simulate different contexts such as home automation, monitoring of critical infrastructures or industrial contexts. In order to validate this approach, legitimate traffic was combined with different malicious/attack traffics targeting the MQTT network. From the raw network traffic generated by MQTTset sensors and cyber-attacks against MQTT, we extracted features necessary to implement a possible detection system. Moreover, in order to validate the dataset, we implemented and compared different machine learning algorithms widely adopted in the cyber-security field (neural network, random forest, naïve bayes, decision tree, gradient boost and multilayer perceptron) by using different balancing approach of the dataset. By comparing balanced and unbalanced dataset, the results reported a high accuracy and F1 score for the unbalanced dataset due to the high number of records for the legitimate traffic that affect the final results. Instead, the balanced dataset reported a low metrics results but a correct distribution of data in the confusion matrices. We learned the importance of a balanced dataset to obtain more realistic results. Finally, basing the evaluation in terms of accuracy and F1 score, results obtained for the considered machine learning algorithms were evaluated, demonstrating how MQTTset can be used for a possible detection system related to the MQTT protocol.

Future work will be related to the application and validation of this dataset to detect attacks against MQTT against an industrial scenario, such as a smart building or an industrial IoT network (Industry 4.0). Based on the obtained results, a future work will be focused on the tuning of the hyperparameters of the machine learning algorithms in order to ensure the accuracy and F1 score about the balanced dataset and to define the best characteristics of the algorithms. Another work may be focused on the extension of MQTTset to integrate other innovative attacks against IoT based protocols and to add more complex data in the scenario, to keep the detection system updated and able to validate/identify threats in real time. Moreover, the dataset will remain totally public so that researchers can use it as a basis to integrate other attacks by extracting their features, implementing their algorithms and combine the dataset with their traffics. In this direction, the dataset could be indeed adopted, for instance, to validate novel threats not included in the dataset, to consider different versions of the MQTT protocol, or to extend MQTTset with different communications protocols or sensor nodes. Moreover, a features selection and statistical analysis approach will be evaluated in order to highlight most relevant features and to train machine learning models with right characteristics. Furthermore, if needed, researches can combine this dataset with other to increase the number of packets and communication traffic. Another interesting future work is to integrate other sensors with different nature (e.g., smart bulbs, smart speakers, etc.) in order to create a more complex ecosystems. Finally, a possible extension on the topic may be directed to the application of the dataset in the field of data analytics, to compare MQTT traffic between networks applied in different contexts (such as medical, smart cities, critical infrastructure) with the MQTT network under analysis. Subsequently, other possible features can be integrated such as encrypted traffic via TLS or using advanced versions of the MQTT protocol, in this case the MQTT 5 version.

## 6. Dataset

The aim of this work is to create a dataset for MQTT available to the research and industrial community to provide a support or starting point for using data analysis techniques or machine learning/artificial intelligence in the IoT context. For this reason, we have decided to make the dataset public and available on the web. The dataset is available at the following address: https://www.kaggle.com/cnrieiit/mqttset.

## Figures and Tables

**Figure 1 sensors-20-06578-f001:**
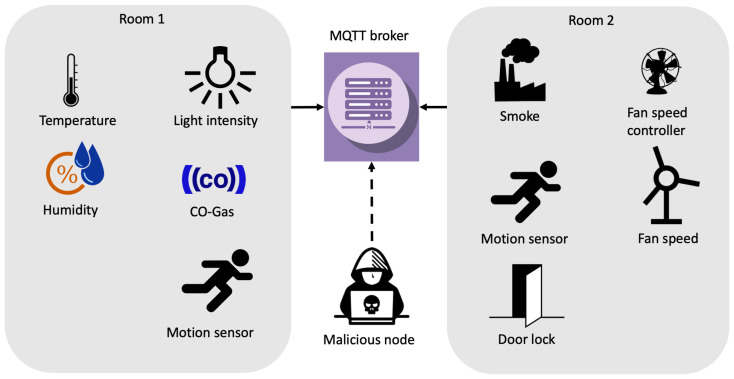
The scenario considered in MQTTset.

**Figure 2 sensors-20-06578-f002:**
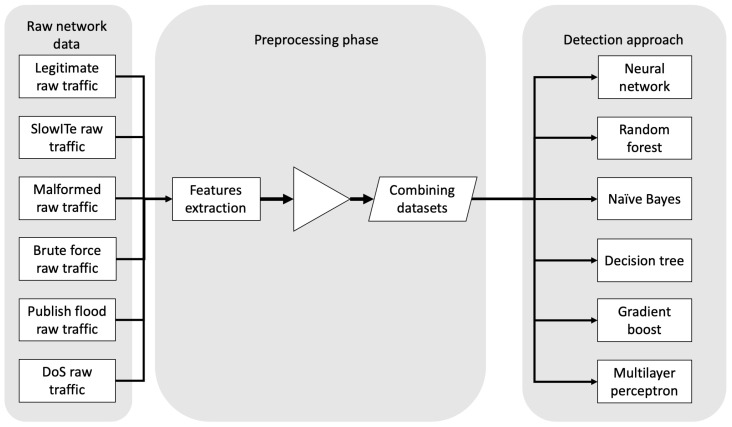
Considered workflow for MQTTset validation.

**Table 1 sensors-20-06578-t001:** Available IoT datasets adopted in detection approach.

Dataset	Lacks
KDDCUP99	Not focused on IoT context
UNSW-NB15	Not focused on IoT context
NIMS	Not focused on IoT context
NLS-KDD	Not focused on IoT context
N-BaIoT	Focused on Wi-Fi communication
IoT-23	Focused on DNS traffic for IoT context
MedBIoT	Authentication phase not found, no MQTT attacks
TON_IoT	Authentication and disconnection phase not found
BoT-IoT	raw PCAP traffic data related to MQTT was not released
Custom datasets	PCAP or raw traffic missing

**Table 2 sensors-20-06578-t002:** IoT sensors adopted in the MQTTset scenario.

Sensor	IP Address	Room	Type	Messages Frequency (s)	Topic	Data Profile
Temperature	192.168.0.151	1	Periodic	60	Temperature	Temperature
Light intensity	192.168.0.150	1	Periodic	1800	Light intensity	Light intensity
Humidity	192.168.0.152	1	Periodic	60	Humidity	Humidity
Motion sensor	192.168.0.154	1	Random	3600	Movement	Movement
CO-Gas	192.168.0.155	1	Random	3600	CO-Gas	CO-Gas
Smoke	192.168.0.180	2	Random	3600	Smoke	Smoke
Fan speed controller	192.168.0.173	2	Periodic	120	Fan speed	Fan speed
Door lock	192.168.0.176	2	Random	3600	Door lock	Door lock
Fan sensor	192.168.0.178	2	Periodic	60	Fan	Fan
Motion sensor	192.168.0.174	2	Random	3600	Movement	Movement

**Table 3 sensors-20-06578-t003:** Attacks executed in the testbed.

Attack	PCAP Size (bytes)	Number of Packets	Time (mm:ss)
flooding denial of service	49,875,539	130,223	05:00
MQTT Publish flood	8,212,656	613	05:00
SlowITe	972,272	9202	10:00
malformed data	1,038,590	10,924	06:00
brute force authentication	1,397,132	14,501	30:00

**Table 4 sensors-20-06578-t004:** The list of extrapolated features.

No	Name	Description	Protocol Layer
1	tcp.flags	TCP flags	TCP
2	tcp.time_delta	Time TCP stream	TCP
3	tcp.len	TCP Segment Len	TCP
4	mqtt.conack.flags	Acknowledge Flags	MQTT
5	mqtt.conack.flags.reserved	Reserved	MQTT
6	mqtt.conack.flags.sp	Session Present	MQTT
7	mqtt.conack.val	Return Code	MQTT
8	mqtt.conflag.cleansess	Clean Session Flag	MQTT
9	mqtt.conflag.passwd	Password Flag	MQTT
10	mqtt.conflag.qos	QoS Level	MQTT
11	mqtt.conflag.reserved	(Reserved)	MQTT
12	mqtt.conflag.retain	Will Retain	MQTT
13	mqtt.conflag.uname	User Name Flag	MQTT
14	mqtt.conflag.willflag	Will Flag	MQTT
15	mqtt.conflags	Connect Flags	MQTT
16	mqtt.dupflag	DUP Flag	MQTT
17	mqtt.hdrflags	Header Flags	MQTT
18	mqtt.kalive	Keep Alive	MQTT
19	mqtt.len	Msg Len	MQTT
20	mqtt.msg	Message	MQTT
21	mqtt.msgid	Message Identifier	MQTT
22	mqtt.msgtype	Message Type	MQTT
23	mqtt.proto_len	Protocol Name Length	MQTT
24	mqtt.protoname	Protocol Name	MQTT
25	mqtt.qos	QoS Level	MQTT
26	mqtt.retain	Retain	MQTT
27	mqtt.sub.qos	Requested QoS	MQTT
28	mqtt.suback.qos	Granted QoS	MQTT
29	mqtt.ver	Version	MQTT
30	mqtt.willmsg	Will Message	MQTT
31	mqtt.willmsg_len	Will Message Length	MQTT
32	mqtt.willtopic	Will Topic	MQTT
33	mqtt.willtopic_len	Will Topic Length	MQTT

**Table 5 sensors-20-06578-t005:** Obtained results from the MQTT dataset.

Algorithm	Accuracy	F1 Score	Training Time (s)	Testing Time (s)
Neural network	0.99326833989724	0.9932468365565741	262.8857	74.2051
Random forest	0.9942991408704308	0.9943007213915611	1375.6648	35.8725
Naïve bayes	0.9879035395431919	0.9897062545007078	45.02647	7.1440
Decision tree	0.9779726992251886	0.9850216439428234	88.7153	1.2932
Gradient boost	0.9911319662528564	0.9916394826795836	1584.3016	10.6267
Multilayer perceptron	0.9468814683726754	0.963694302875892	3024.1888	18.4380

**Table 6 sensors-20-06578-t006:** Confusion matrix neural network.

				Predicted			
		**Bruteforce**	**DoS**	**Flood**	**Legitimate**	**Malformed**	**SlowITe**
**Actual**	Bruteforce	3198	167	0	559	2	425
	DoS	287	28,050	0	10,740	0	0
	Flood	3	17	29	124	11	0
	Legitimate	6443	83	0	3,568,146	43	0
	Malformed	2226	77	0	566	29	380
	SlowITe	1437	0	0	802	6	516

**Table 7 sensors-20-06578-t007:** Confusion matrix random forest.

				Predicted			
		**Bruteforce**	**DoS**	**Flood**	**Legitimate**	**Malformed**	**SlowITe**
**Actual**	Bruteforce	3195	375	0	212	560	9
	DoS	191	32,812	0	5997	76	1
	Flood	1	0	89	93	1	0
	Legitimate	1632	7278	0	3,565,038	712	55
	Malformed	943	142	0	427	1757	9
	Slowite	150	360	0	1136	302	813

**Table 8 sensors-20-06578-t008:** Confusion matrix naïve bayes.

				Predicted			
		**Bruteforce**	**DoS**	**Flood**	**Legitimate**	**Malformed**	**SlowITe**
**Actual**	Bruteforce	961	40	10	3411	9	20
	DoS	112	27,866	0	11,004	0	95
	Flood	2	0	88	92	1	1
	Legitimate	12,051	0	0	3,550,861	0	11,803
	Malformed	44	90	27	2449	443	225
	Slowite	17	0	0	2331	8	405

**Table 9 sensors-20-06578-t009:** Confusion matrix decision tree.

				Predicted			
		**Bruteforce**	**DoS**	**Flood**	**Legitimate**	**Malformed**	**SlowITe**
**Actual**	Bruteforce	3286	358	0	211	496	0
	DoS	229	32,798	0	5992	57	1
	Flood	1	0	89	93	1	0
	Legitimate	52,541	8967	0	3,505,850	7290	67
	Malformed	1013	140	1	426	1695	3
	Slowite	193	371	0	1091	293	813

**Table 10 sensors-20-06578-t010:** Confusion matrix gradient boost.

				Predicted			
		**Bruteforce**	**DoS**	**Flood**	**Legitimate**	**Malformed**	**SlowITe**
**Actual**	Bruteforce	1787	67	506	1579	411	1
	DoS	29	15,744	9666	6403	0	7235
	Flood	2	0	0	182	0	0
	Legitimate	20	1385	0	3,573,310	0	0
	Malformed	557	18	308	1393	987	15
	Slowite	123	0	398	1720	123	397

**Table 11 sensors-20-06578-t011:** Confusion matrix multilayer perceptron.

				Predicted			
		**Bruteforce**	**DoS**	**Flood**	**Legitimate**	**Malformed**	**SlowITe**
**Actual**	Bruteforce	3568	209	5	179	49	341
	DoS	391	32,874	0	5788	12	12
	Flood	11	11	56	91	15	0
	Legitimate	329	175,806	0	3,394,175	3817	588
	Malformed	2121	165	5	460	277	250
	Slowite	1045	10	0	799	12	895

**Table 12 sensors-20-06578-t012:** Obtained results from the MQTT dataset with balanced dataset.

Algorithm	Accuracy	F1 Score	Training Time (s)	Testing Time (s)
Neural network	0.9044728333333333	0.9023636467243322	778.1805	144.2180
Random forest	0.9159708333333333	0.9140355032443288	2298.2762	125.8504
Naïve bayes	0.643889	0.6872843841719165	85.2840	13.7836
Decision tree	0.9159608333333333	0.9140241688909468	148.8115	2.3031
Gradient boost	0.8795693333333333	0.8727044450930602	8840.0049	18.1375
Multilayer perceptron	0.9038521666666667	0.9018922771095824	5714.4811	27.2843

**Table 13 sensors-20-06578-t013:** Confusion matrix neural network with augmented and balanced traffic.

				Predicted			
		**Bruteforce**	**DoS**	**Flood**	**Legitimate**	**Malformed**	**SlowITe**
**Actual**	Bruteforce	500,368	26,869	1369	1158	70,236	0
	DoS	8816	525,989	6494	49,950	8751	0
	Flood	1200	4800	457,200	133,200	3600	0
	Legitimate	324	13,019	37,366	2,948,060	1231	0
	Malformed	175,020	20,280	1680	7800	395,220	0
	Slowite	0	0	0	0	0	600,000

**Table 14 sensors-20-06578-t014:** Confusion matrix random forest with augmented and balanced traffic.

				Predicted			
		**Bruteforce**	**DoS**	**Flood**	**Legitimate**	**Malformed**	**SlowITe**
**Actual**	Bruteforce	529,389	25,547	769	1158	43,137	0
	DoS	7797	523,910	7640	50,153	10,500	0
	Flood	1200	3600	453,600	140,400	1200	0
	Legitimate	340	11,707	28,546	2,957,346	2061	0
	Malformed	142,380	15,300	2760	8040	431,520	0
	Slowite	0	0	0	0	0	600,000

**Table 15 sensors-20-06578-t015:** Confusion matrix naïve bayes with augmented and balanced traffic.

				Predicted			
		**Bruteforce**	**DoS**	**Flood**	**Legitimate**	**Malformed**	**SlowITe**
**Actual**	Bruteforce	591,126	0	0	8874	0	0
	DoS	169,475	430,045	70	250	160	0
	Flood	300,000	0	292,800	6000	0	1200
	Legitimate	1,106,594	0	28,643	1,864,763	0	0
	Malformed	477,060	16,620	7740	13,980	84,600	0
	Slowite	0	0	0	0	0	600,000

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
