# Peer review of "MQTTset, a New Dataset for Machine Learning Techniques on MQTT"

_sensors, 2020, doi:10.3390/s20226578_

Round 1

Reviewer 1 Report

This paper presents a data set for MQTT smart home sensor data.

While it only includes 8 sensors, it does have some security attack data generated from implementations of 5 typical attack mechanisms.

Also, a simple analyis of the data based on off-the-shelf data analyis techniques is performed to detect the attacks based on training and learning.
These techniques sow very high accuracy of detection, all abov 98%

Overall, I consider this direction as useful to validate different techniques for security analysis for IoT systems.

Yet, this paper has a bit limited value due to the following

- data seems to be very easy to analyse, all techniques give very good results

- only 8 sensors. This may also be the reason for the above.

There is an overview of existing datasets, but this could be more structured, e.g. in a table. It is stated at the end of section 4 that this is hence a "good starting point". Here, I would disagree - it looks a bit too simple

Also, there is little discussion if existing techniques could be used to detect attacks (which are not specific to IoT and MQTT).

Overall, I would recommend to either do a larger / more complex data set which poses more challenges.
Or, to provide a toolbox such that others can create such more challenging sets.

Reviewer 2 Report

The paper creates and publishes an IoT dataset for security research.

The study has a solid contribution as the number of datasets is scarce in this area. However, the study suffers from various points:

  • The literature review is not complete. There are various IoT dataset studies that have not been covered in the literature review:
    • https://www.stratosphereips.org/datasets-iot23
    • https://www.researchgate.net/publication/338765489_MedBIoT_Generation_of_an_IoT_Botnet_Dataset_in_a_Medium-sized_IoT_Network
    • https://archive.ics.uci.edu/ml/datasets/detection_of_IoT_botnet_attacks_N_BaIoT

            Based on the more detailed literature review, the study should underline the contribution of this study.

  • The details of the system topology that is used for data generation are not given. Network segments, the location of other components (like a firewall), the location of attackers, the sniffing points should be demonstrated and explained well. The physical locations of the sensor can be also shown in the figure.

  • The crucial part of the data generation is the simulation of normal system usage and labeling of normal and attack records. Which assurance can be given so that the dataset reflects a normal life pattern? For instance, what are the periodicity values of sensors or how random actions are created in IoT networks (more details are needed especially for “type” variable in Table 1.) The labeling of the data should be explained clearly.

  • The attack types include patterns that rely on heavy traffic (i.e., denial of service attacks or brute force attempts). For instance, attacks covering the remote exploitation of the devices have not been considered. It can be argued that the attacks simulated in the study are very easy to detect, considering the periodicity and other properties of the IoT traffic. The paper requires more discussion about the attack type selection and, thus, the limitation of the dataset.

  • The reasons for selecting such features should be elaborated more. For instance, why did the authors not select host-based or host-to-host high-level statistics that may easily grasp the essence of traffic profiles?

  • One of the important aspects of datasets is the balance of malicious and normal traffic. Can the dataset be considered as balanced or unbalanced? Such a discussion is needed.

  • Although the main contribution of this paper is the dataset itself, the analytical part of the study is so weak.
    • The hyperparameters and other details of the machine learning models are not given in the experimentation part.
    • It is not clear if the execution time is related to training or testing time.
    • As this study introduces the dataset, the statistical properties of the features could be explained more.
    • The analytical part could be extended with the features which have more discrimination value.

  • A discussion about the misclassified instances can be given. Which attack categories have not been detected so well? A detailed confusion matrix analysis can help in that sense.

  • There is no discussion about the lessons learned regarding the data creation endeavour. It could be so valuable to discuss them and inform other researchers.

All in all, the study requires a major update on the aforementioned issues to be published as a journal publication. 

Round 2

Reviewer 2 Report

The paper has gone through an important update. I think this version has more analytical results and better explanations about the dataset.

However, still, the following issues would be solved before delivering the final version:

+ Labels of Confusion matrix dimensions. Which axis is the detection result and which one is the actual label?

+ Although this paper does not aim to optimize the hyperparameters of the ML algorithms, it is important to report the parameters of the used algorithms for the sake of repeatability.
